# Aptamer based proteomic pilot study reveals a urine signature indicative of pediatric urinary tract infections

Liang Dong[1], Joshua Watson[2], Sha Cao[3], Samuel Arregui[1], Vijay Saxena[1], John Ketz[4], Abduselam K. Awol[5], Daniel M. Cohen[6], Jeffrey M. Caterino[7], David S. Hains[1], Andrew L. Schwaderer[1] *

1 Department of Pediatrics, Indiana University, Indianapolis, Indiana, United States of America, 2 Division of Infectious Disease, Nationwide Children's Hospital, Columbus, Ohio, United States of America, 3 Department of Biostatistics, Indiana University, Indianapolis, Indiana, United States of America, 4 The Research Institute at Nationwide Children's Hospital, Columbus, Ohio, United States of America, 5 Earlham College, Richmond, Indiana, United States of America, 6 Division of Emergency Medicine, Nationwide Children's Hospital, Columbus, Ohio, United States of America, 7 Division of Emergency Medicine, The Ohio State University, Columbus, Ohio, United States of America

* schwadea@iu.edu

**Data Availability Statement:** All relevant data are within the paper and Supporting Information files.

**Funding:** Eli Lily foundation award (https://www.lilly.com/who-we-are/lilly-foundation) ALS

## Abstract

### Objective

Current urinary tract infection (UTI) diagnostic strategies that rely on leukocyte esterase have limited accuracy. We performed an aptamer-based proteomics pilot study to identify urine protein levels that could differentiate a culture proven UTI from culture negative samples, regardless of pyuria status.

### Methods

We analyzed urine from 16 children with UTIs, 8 children with culture negative pyuria and 8 children with negative urine culture and no pyuria. The urine levels of 1,310 proteins were quantified using the Somascan™ platform and normalized to urine creatinine. Machine learning with support vector machine (SVM)-based feature selection was performed to determine the combination of urine biomarkers that optimized diagnostic accuracy.

### Results

Eight candidate urine protein biomarkers met filtering criteria. B-cell lymphoma protein, C-X-C motif chemokine 6, C-X-C motif chemokine 13, cathepsin S, heat shock 70kDA protein 1A, mitogen activated protein kinase, protein E7 HPV18 and transgelin. AUCs ranged from 0.91 to 0.95. The best prediction was achieved by the SVMs with radial basis function kernel.

### Conclusions

Biomarkers panel can be identified by the emerging technologies of aptamer-based proteomics and machine learning that offer the potential to increase UTI diagnostic accuracy, thereby limiting unneeded antibiotics.

and DSH The Research Institute at Nationwide Hospital intramural funds (https://www.nationwidechildrens.org/research). ALS National Institute on Aging (https://www.nia.nih.gov) National Institute of Diabetes and Digestive and Kidney Diseases (https://www.niddk.nih.gov) Grant numbers: R01AF050801 (salary support for JC, ALS and DSH), R01DK106286 (salary support for ALS and DSH) and R01DK117934 (salary support for ALS and DSH). The sponsors played no role in study design, data collection and analysis, decision to publish, or preparation of the manuscript.

**Competing interests:** I (Andrew Schwaderer) have consulted for Allena Pharmaceuticals on a topic unrelated to this manuscript; This does not alter our adherence to PLOS ONE policies on sharing data and materials. Otherwise there are no conflicts to report. The work in this manuscript is original, not previously published, and not submitted for publication or consideration elsewhere.

# Background

Urinary tract infections (UTIs) are frequently encountered. UTIs account for 7 million office visits and 400,000 hospitalizations annually in the United States [1, 2]. The aforementioned hospitalizations for UTIs increased 52% between 1998 and 2011 and resulted in an estimated 2.8 billion dollars of cost. In children, UTIs account for 7% of emergency department (ED) antibiotic prescriptions [3]. Unfortunately, antibiotic resistance in uropathogens is increasing [4]. The diagnosis of UTIs is typically made at the point of care by symptoms and the identification of nitrites and /or leukocyte esterase (LE) on urinalysis (UA) and/or urine dipstick [5]. Ultimately urine culture results with 50,000 colony forming units (cfu)/ml of a uropathogen is used to confirm a clinical UTI [5]. However accurate urine culture results can be dependent on proper collection methodology and take 24–48 hours to complete [6]. Further, the use of LE on urine dipsticks has limitations including limited sensitivity and specificity [5]. Specifically non UTI, infectious and/or inflammatory conditions such chlamydia, appendicitis and interstitial nephritis can result in positive urine leukocyte esterase and negative urine cultures [7].

A timely and accurate UTI diagnosis is important in clinical care. Initiating antibiotics in a patient with suspected UTI that is actually culture negative pyuria exposes the patient to unneeded antibiotics and potentially increases the risk of antibiotic resistant bacteria [8]. Conversely, waiting for culture results before treating a patient who has a true UTI risks complication from disease progression from cystitis to pyelonephritis or even urosepsis [9]. Methodologies that increase accuracy of UTI diagnosis are needed.

The discoveries of new biomarkers are fundamental to improving clinical care. Several challenges have limited protein-based biomarker discovery with traditional antibody based ELISAs. Specifically, ELISAs are time consuming to perform and the required antibodies have inherent costs, instability, batch to batch variation, storage requirements limited dynamic ranges and are difficult to multiplex [10–12]. SOMAscan™ (Somologics, Boulder CO) uses SOMAmer (slow-off-rate modified aptamer) protein binding reagents. Aptamers are modified DNA with high affinity ($10^9$–$10^{12}$ M) and high specificity for their cognate analytes comparable to sandwich ELISA performance that have been used for biomarker discovery [13]. For example, in 2018 the Somascan aptamer-based platform was utilized to discover unique protein profiles in autoimmune cholangitis [14]. Aptamers are being explored as affordable, sensitive, specific, user-friendly point of care tests on a variety of platforms [15]. Aptamers have the ability to perform in formats where antibodies often perform poorly such as homogenous multiplex assays, do not degrade when stored at room temperature as a dry lyophilized reagent at room temperature and have minimal to no batch to batch variation [12]. Thus there is speculation that aptamers may replace antibodies in future diagnostics [16]. The objectives of this pilot study are to determine if SOMAscan$^M$ aptamer-based comparison of (a) children with neither pyuria nor growth on culture, (b) children with pyuria but no growth on urine culture and (c) children with pyuria and 50,000 cfu/ml of *E.coli* on urine culture will reveal a protein profile unique to children with UTI along with performing an initial experiment comparing aptamer based detection with antibody based detection using an adult emergency cohort for the latter.

# Methods

## Study approval and patients

The study was approved and granted a waiver of informed consent by the Nationwide Children's Hospital Institutional review board (IRB13-00090). Samples were prospectively obtained in the ED and main campus Urgent Care at Nationwide Children's Hospital, Columbus, Ohio as previously described [17]. Inclusion criteria consisted of dipstick urinalysis and urine culture

performed for any clinical indication and availability of excess urine sample. Children who had received antibiotic treatment within 7 days before ED presentation were excluded. Samples from a second cohort were collected from the Ohio State University Wexner Medical Center Emergency Department (ED) patients. Institutional review board approval was obtained and informed consent was obtained from each patient or appropriate proxy. Inclusion criteria consisted of ED $\geq$ 65 years of age that had a urinalysis for suspected UTI. Exclusion criteria consisted of chronic intermittent catheterization, UTI or positive urine culture or genitourinary procedure in the prior 30 days, antibiotic use in the past 14 days, immunosuppression, hemodialysis, homelessness, previous enrollment, incarceration, non-English speaking, trauma team activation, and lack of patient or proxy ability to give consent or respond to the survey.

## Sample collection and processing

We prospectively collected urine samples as previously described [17]. Samples were collected when a research assistant was available to collect samples resulting in a convenience collection. After ensuring sufficient urine volume was available for clinical diagnostic tests, excess urine was immediately collected in AssayAssure urine collection tubes (Thermo Scientific, Waltham, MA) containing a bacteriostatic preservative that suppresses nuclease and protease activity and preserves urine specimens at room temperature for up to 26 days per the manufacturer. We previously independently confirmed protein stability for 14 days [17]. Samples were processed within 7 days of collection and centrifuged at 3,000 rpm for 5 minutes with the supernatant saved in 300 to 500 μl aliquots then stored at −80˚C until use.

## Sample selection and groups

Samples were selected from our previously reported cohort of pediatric ED patients that had sufficient urine volume for Somalogic analysis, were collected by clean catch and fit criteria for the following groups: (a) *UTI* defined by $\geq$1+ LE on urine dipstick and $\geq$50,000 cfu/ml of *E. coli* on urine culture; (b) *Culture negative (CN) pyuria* defined by $\geq$1+ LE on urine dipstick and no growth on urine culture and (d) *CN no pyuria* defined by negative LE on urine dipstick and no growth on urine culture [5, 17, 18]. The UTI group was further divided between those with and without fevers $\geq$ 100.4˚ Fahrenheit/38˚ Celsius (either measured in the ED or by report at home). Urine culture functioned as the "reference standard". The adult ED cohort was divided into a culture negative and culture positive group.

## Urine aptamer proteomic evaluation

One aliquot of the selected samples were sent on dry ice to Somalogic Inc. (Boulder, CO) to measure concentrations of 1,310 proteins via SOMAscan™ analysis. The SOMAscan™ results were presented as relative fluorescent units (RFU) per ml. Urine protein levels were normalized to urine creatinine which were measured using the Oxford colorimetric assay (Oxford Biomedical Research, Oxford, MI).

## Urine antibody based protein detection

A V-PLEX Human GM-CSF Kit (Mesoscale Discovery, Rockville, MD, catalog # K151RID-1) was used for validation, in an adult ED cohort, of urine granulocyte-macrophage colony stimulating factor (GM-CSF) levels according to the manufacturer's directions. We chose the mesoscale array because it has a large dynamic range of detection (0.14–770 pg/ml), is labeled for use in urine and we have experience with the assay [19]. Results were normalized to urine creatinine as described previously for the urine aptamer levels.

## Statistical analysis

Demographics and presenting symptoms were compared with Graphpad Prism (La Jolla, CA) using the chi square test if percentages or proportions were evaluated and the t-test if continuous data was evaluated. Groups were compared by the Wilcoxon test with SPSS software (IBM corporation, Armonk NY). Proteins were filtered by meeting all of the following criteria: (a) significantly different levels between the UTI group (febrile and afebrile combined) versus the CN-pyuria group; (b) significantly different levels between the UTI group (febrile and afebrile combined) vs the CN-no pyuria group and (c) but not significantly different levels between the CN-pyuria group vs CN-no pyuria group. Significance was assigned for a p value of $< 0.05$ and the results were further filtered for a p value $< 0.01$. Next, proteins with that had a p-value $< 0.01$ and under curve (AUC) of $> 0.9$ were selected as candidate biomarkers. A general guide for interpreting the utility of a biomarker based on AUC is: "fail = 0.5–0.6", "poor" = 0.6–0.7, "fair" = "0.7–0.8, "good" = 0.8–0.9 and "excellent" = 0.9–1.0 [20]. AUCs and the concentration with likelihood ratios (LRs) were calculated using Graphpad Prism.

## Support vector machine (SVM) predictive model optimization

Feature selection plays a crucial role in biomedical data mining. [21] Three different feature selection approaches were considered to reduce the data dimensionality before the model was trained on training subset in each fold of inner leave-one-out cross-validation: (a) feature selection based on the Wilcoxon rank sum test to screen proteins with expression strongly associated with UTI. (b) Feature ranking on the basis of random forest feature importance scores computed from the Gini impurity reduction. (c) ReliefF feature selection techniques [22]. Given our sample size, we performed hyperparameter tuning and model optimization using leave-one-out cross-validation in an inner loop. We conducted a grid search to explore the optimal hyperparameter space including a range of values for gamma and/or C for support vector classifiers with either linear or RBF kernel. The accuracy was calculated at each cross-validation split on the validation set. The mean accuracy was used as a metric for model selection. To assess the predictive performance, we further computed the performance estimates of our models on unseen data (test set) using 5-fold cross-validation in the outer loop. The overall unbiased generalization performance of the optimal model was evaluated by the mean AUC values of the receiver operating characteristic (ROC) curve, obtained in each iteration of the cross-validation split. The class probability estimate of each sample was calculated based on decision values of SVM using the parameters learned in Platt scaling [23]. A number of Python libraries and R packages were used in data analysis and machine learning processes including Pandas, Scikit-Learn, skrebate, ggplot2, dplyr, ROCR, and pROC. A schematic of the methodology for feature selection and nested cross validation is presented as S2 Material.

## Figure generation

Figures were generated using Graphpad Prism, Microsoft Powerpoint (Microsoft corporation, Redmond, WA) or by web-based Lucidchart tool (https://www.lucidchart.com).

## Results

### Patients

We included urine samples from 32 patients (4 males and 28 females) with a median age of 7.1 years (interquartile range, 4.7–14.0). Sixteen patients met criteria for UTI group, 8 patients had CN-pyuria and 8 patients had CN-no pyuria. The UTI group was evenly divided between those with and without fevers. No patients were immunosuppressed. Two patients in the UTI

**Table 1. Epidemiology and presenting symptoms^ of groups.**

| | UTI (n = 16) | CN pyuria (n = 8) | CN no pyuria (n = 8) | P value |
|---|---|---|---|---|
| Mean age (years) ± std dev | 8.2 ± 4.7 | 11.4 ± 5.9 | 7.2 ± 4.2 | 0.217 |
| Female:male (% female) | 15:1 (94%) | 7:1 (88%) | 5:3 (63%) | 0.133 |
| Fever | 8 (50%) | 2 (25%) | 0 (0%) | *0.041 |
| Dysuria | 5 (31%) | 2 (25%) | 3 (38%) | 0.793 |
| Frequency | 4 (25%) | 1 (13%) | 0 (0%) | 0.272 |
| Urgency/enuresis | 4 (25%) | 2 (25) | 1 (13%) | 0.760 |
| Suprapubic pain | 4 (25%) | 1 (13%) | 0 (0%) | 0.272 |
| Abdominal pain | 8 (50%) | 4 (50%) | 3 (38%) | 0.828 |
| Back/flank pain | 2 (13%) | 3 (38%) | 0 (0%) | 0.105 |

^other presenting symptoms included syncope (1 in UTI group and one in normal U urine group, "bump" on testicle (1 in CN no pyuria group), headache (one in normal urine group), foul smelling urine (1 in normal urine group) and memory loss (1 in CN pyuria group)

*statistically significant, $p < 0.05$ with the significant difference between UTI and CN no pyuria group.

group had a history of kidney stones. One 5-year old patient in the UTI group had a history of congenital hydronephrosis. There were no statistically significant differences in age, sex or presenting symptoms among groups, with the exception of a higher percentage of patients with fever in the UTI compared to the CN no pyuria group (Table 1).

## Identification of proteins elevated during UTI

We identified 133 proteins that were significantly elevated (p value < 0.05) in UTI vs. the culture negative pyuria comparison and) UTI vs. the CN no pyuria group but were not statistically different when the CN pyuria group was compared to the CN no pyuria urine group (S3 Material). To focus on the most differentiating proteins between groups, we filtered for a p value < 0.01 and identified 32 proteins that were elevated in the UTI group, but not the CN-pyuria or CN no pyuria groups (Table 2). The candidates that meet the p value < 0.01 criteria were filtered for AUC curves > 0.9 to determine candidate proteins as "excellent" biomarkers to differentiate culture positive (febrile + afebrile UTI) samples from the combined culture negative samples (CN-pyuria and CN-no pyuria) with the results presented in Fig 1. Scatterplots of the urine biomarker to creatinine ratio in each group along with the urine biomarker to creatinine ratio threshold ("cut off" levels, sensitivity, specificity and likelihood ratios to differentiate UTI (febrile + afebrile) samples from the control samples (CN no pyuria + CN pyuria) are presented in Fig 2.

## Antibody based protein detection

Antibody based protein detection was performed on GM-CSF in the adult patients enrolled from the Ohio State University Wexner Medical Center ED, with the results divided into culture negative (n = 10) and culture positive (n = 6). GM-CSF was selected from Table 1 because it could be tested in a commercially available V-PLEX assay and has previously reported relevance in UTI pathophysiology. The Mesoscale V-PLEX antibody-based protein detection had an AUC of 0.9333, comparable to the AUC of 0.8906 with the Nationwide Children's ED cohort and Somascan™ aptamer-based measurement (Fig 3).

## Support vector machine (SVM) predictive model optimization

The Random forest, ReliefF, and Wilcoxon rank-sum test were applied in feature selection to determine the best combination of urine protein biomarkers that achieved the best prediction

**Table 2. Urine biomarker levels (urine biomarker (RFU)/ urine creatinine (mg)).**

| Protein | Median CN no pyuria | Median CN pyuria | Median UTI | p value | | |
| --- | --- | --- | --- | --- | --- | --- |
| | | | | UTI vs CN-no pyuria | UTI vs CN -pyuria | CN-no pyuria vs CN-pyuria |
| Alpha-2-macroglobulin | 812 ± 1055 | 1170 ± 2104 | 15462 ± 153990 | 0.001 | 0.006 | 0.442 |
| B-cell lymphoma 6 protein | 678 ± 1132 | 542 ± 27213 | 46537 ± 656650 | <0.001 | 0.006 | 0.721 |
| BH3-interacting domain death agonist | 341 ± 1900 | 630 ± 263 | 33323 ± 6533 | 0.006 | <0.001 | 0.234 |
| C-X-C motif chemokine 11 | 55.12 ± 439 | 3604 ± 548 | 1331 ± 95541 | 0.005 | 0.001 | 0.328 |
| C-X-C motif chemokine 13 | 44 ± 972 | 80 ± 163 | 474 ± 8735 | <0.001 | 0.003 | 0.195 |
| C-X-C motif chemokine 6 | 185 ± 942 | 145 ± 131 | 3342 ± 35842 | 0.001 | <0.001 | 0.645 |
| Calcium/calmodulin-dependent protein kinase type 1 | 1452 ± 3393 | 2126 ± 1951 | 8080 ± 18809 | 0.004 | 0.009 | 0.328 |
| Cathepsin S | 114 ± 645 | 314 ± 255 | 2513 ± 33882 | <0.001 | <0.001 | 0.195 |
| Endothelial monocyte-activating polypeptide 2 | 277 ± 698 | 394 ± 295 | 1788 ± 5360 | 0.002 | 0.006 | 0.382 |
| Granulocyte-macrophage colony-stimulating factor | 44 ± 111 | 46 ± 40 | 296 ± 930 | 0.004 | <0.001 | 0.959 |
| Gro-beta/gamma | 162 ±843 | 377 ± 259 | 6564 ± 93973 | 0.002 | 0.001 | 0.161 |
| Growth-regulated alpha protein | 287 ±1952 | 1490 ± 1675 | 27124 ± 146868 | <0.001 | 0.002 | 0.195 |
| Heat shock 70 kDa protein 1A | 244 ± 1044 | 1190 ± 1379 | 12440 ± 78706 | <0.001 | 0.003 | 0.083 |
| Heat shock cognate 71 kDa protein | 5838 ± 30168 | 18610 ±13963 | 69018 ± 160844 | 0.002 | 0.003 | 0.382 |
| Histone H2A type 3 | 5148 ± 7319 | 6517 ± 42825 | 56331 ± 130985 | 0.001 | 0.005 | 0.574 |
| Immunoglobulin A | 39421 ±73514 | 30451 ± 67150 | 222417 ± 221679 | 0.009 | 0.007 | 0.878 |
| Interstitial collagenase | 55.8 ± 1.30 | 94 ± 5270 | 890 ± 33547 | <0.001 | 0.009 | 0.130 |
| Macrophage-capping protein | 1917 ± 6597 | 1899 ± 1555 | 33978 ± 107421 | 0.004 | 0.001 | 0.878 |
| Mitogen-activated protein kinase 9 | 371 ± 478 | 346 ± 21489 | 55799 ± 136700 | <0.001 | 0.001 | 0.574 |
| Mothers against decapentaplegic homolog 3 | 164 ± 535 | 204 ± 189 | 1162 ± 1576 | 0.009 | 0.004 | 0.645 |
| Nucleoside diphosphate kinase A | 1850 ± 10050 | 3016 ± 4232 | 45384 ± 429.0 | 0.003 | 0.007 | 0.645 |
| Proteasome activator complex subunit 1 | 1800 ± 7904 | 1590 ± 1585 | 27296 ± 74459 | 0.005 | 0.001 | 0.798 |
| Proteasome activator complex subunit 3 | 54 ± 57 | 73 ± 244 | 773 ± 2986 | 0.001 | 0.007 | 0.382 |
| Protein E7_HPV18 | 96 ± 131 | 247 ± 6253 | 17510 ± 51559 | <0.001 | 0.001 | 0.195 |
| Pulmonary surfactant-associated protein D | 9529 ± 45588 | 15407 ± 33968 | 185604 ± 371714 | 0.005 | 0.007 | 1.000 |
| Ras GTPase-activating protein 1 | 55 ± 225 | 74 ± 49 | 486 ± 1565 | 0.004 | <0.001 | 0.721 |
| Small nuclear ribonucleoprotein F | 141 ± 295 | 336 ± 183 | 1155 ± 9453 | 0.002 | 0.007 | 0.279 |
| Stress-induced-phosphoprotein 1 | 1331 ± 15285 | 1889 ±3156 | 31714 ± 142272 | 0.005 | 0.001 | 0.721 |
| Tissue-type plasminogen activator | 1828 ± 1290 | 360 ± 724 | 2326 ± 15039 | 0.004 | 0.004 | 0.645 |
| Transgelin-2 | 3383 ± 10436 | 6614 ± 5343 | 50796 ± 135169 | 0.001 | 0.001 | 0.645 |
| Tumor necrosis factor receptor superfamily member 13C | 180 ± 593 | 324 ± 242 | 1649 ± 3945 | 0.002 | 0.001 | 0.382 |
| Ubiquitin carboxyl-terminal hydrolase isozyme L1 | 791 ±1828 | 991 ± 676 | 76223 ± 38825 | 0.004 | 0.001 | 0.959 |

performance. A total of 45 most frequently occurring urine proteins were selected during the feature selection process, with 29% of them overlapping with each other. As shown by the Venn diagram, the overlapped urine protein biomarkers selected by at least two methods include *MAPK9, CXCL1, CXCL6, CXCL13, HSPA1A, E7, TYK2, PAME3, BCL6, LTF, HIST3H2A and SUMO3* (Fig 3).

The best AUC score was achieved with the SVM classifier with a radial basis function kernel (AUC score of 0.91). SVM worked best with the dataset consisting of Random forest algorithm selected urine proteins. The thirteen most frequently occurring proteins identified in feature

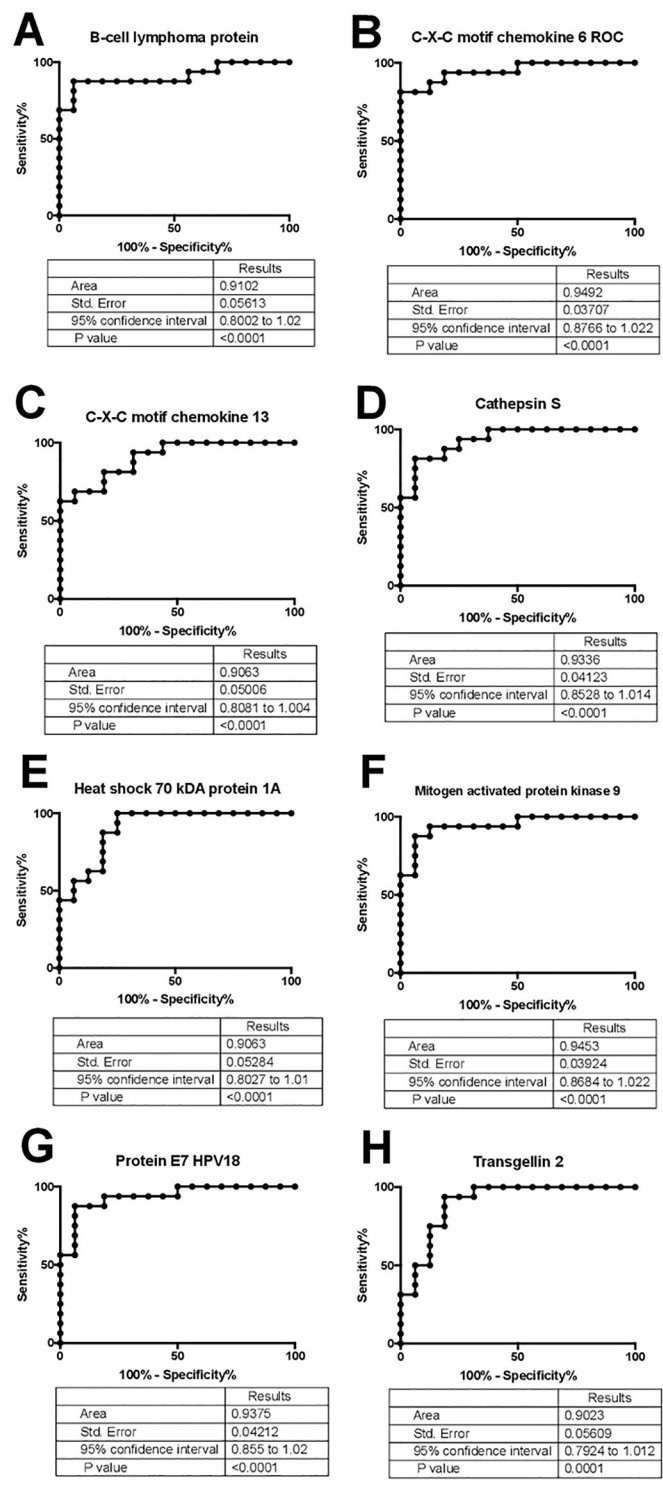

**Fig 1. Candidate biomarker ROCs: Area under the curve (AUCs) demonstrating the 8 candidate biomarkers that meet p value filtering criteria and had AUCs > 0.9.** B-cell lymphoma protein (**A**), C-X-C motif chemokine 6 (**B**) C-X-C motif chemokine 13 (**C**), Cathepsin S (**D**), Heat shock 79kDA protein 1A (**E**), Mitogen activated protein kinase (**F**), Protein E7 HPV18 (**G**) and Transgellin 2 (**H**) are presented.

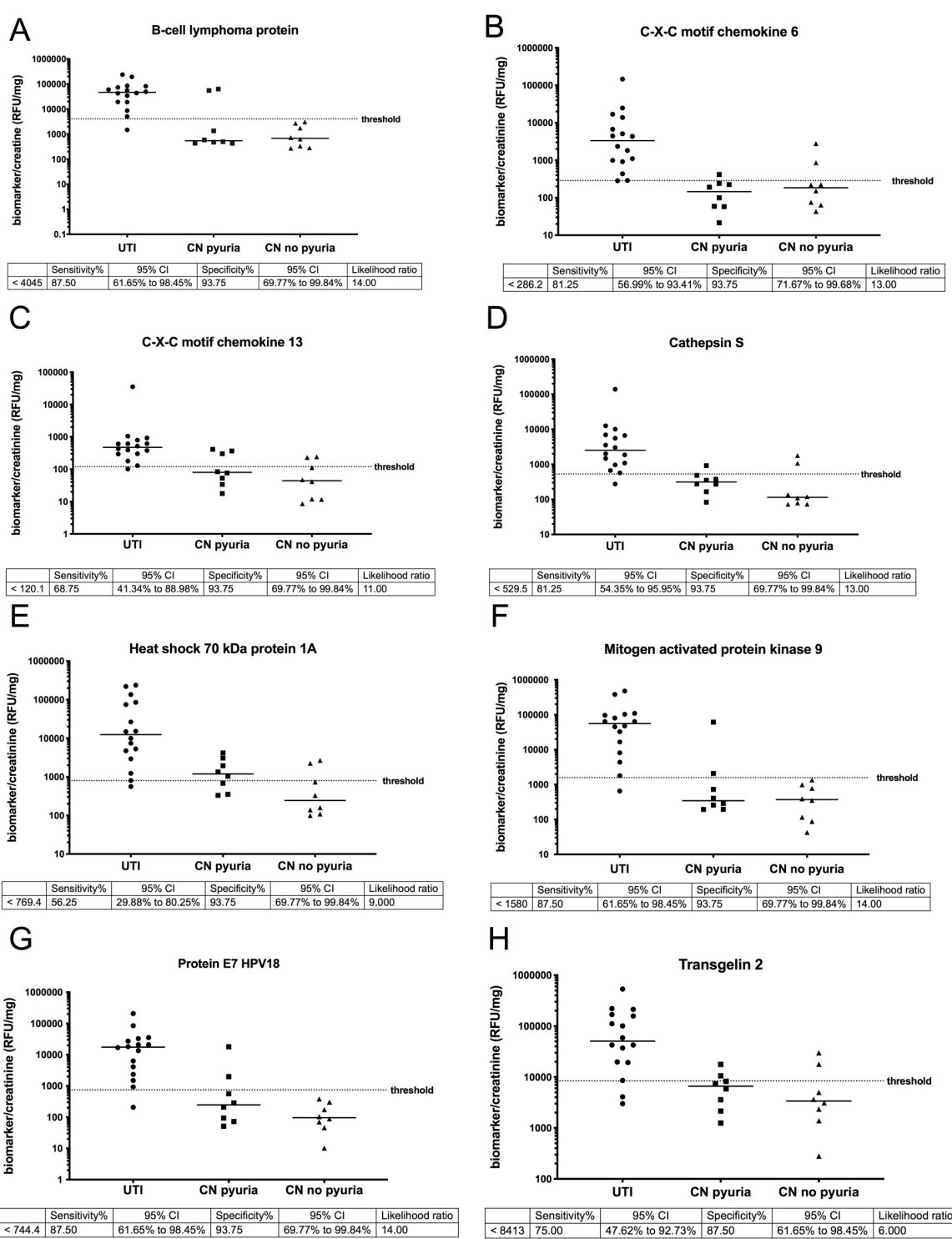

**Fig 2. Candidate biomarkers scatterplots: Scatter plots of urine biomarkers that met p value and AUC criteria are presented to show threshold values that differentiate between UTI and no UTI (CN pyuria and CN no pyuria urine).** The CN pyuria and CN no pyuria samples were separated for graphical, but not for determination of the likelihood ratio (LR). Threshold levels and LRs are presented for B-cell lymphoma protein (**A**), C-X-C motif chemokine 6 (**B**) C-X-C motif chemokine 13 (**C**), Cathepsin S (**D**), Heat shock 79kDA protein 1A (**E**), Mitogen activated protein kinase (**F**), Protein E7 HPV18 (**G**) and Transgelin 2 (**H**) Y axis units are presented as relative fluorescent units (RFU) of biomarker/ mg creatinine.

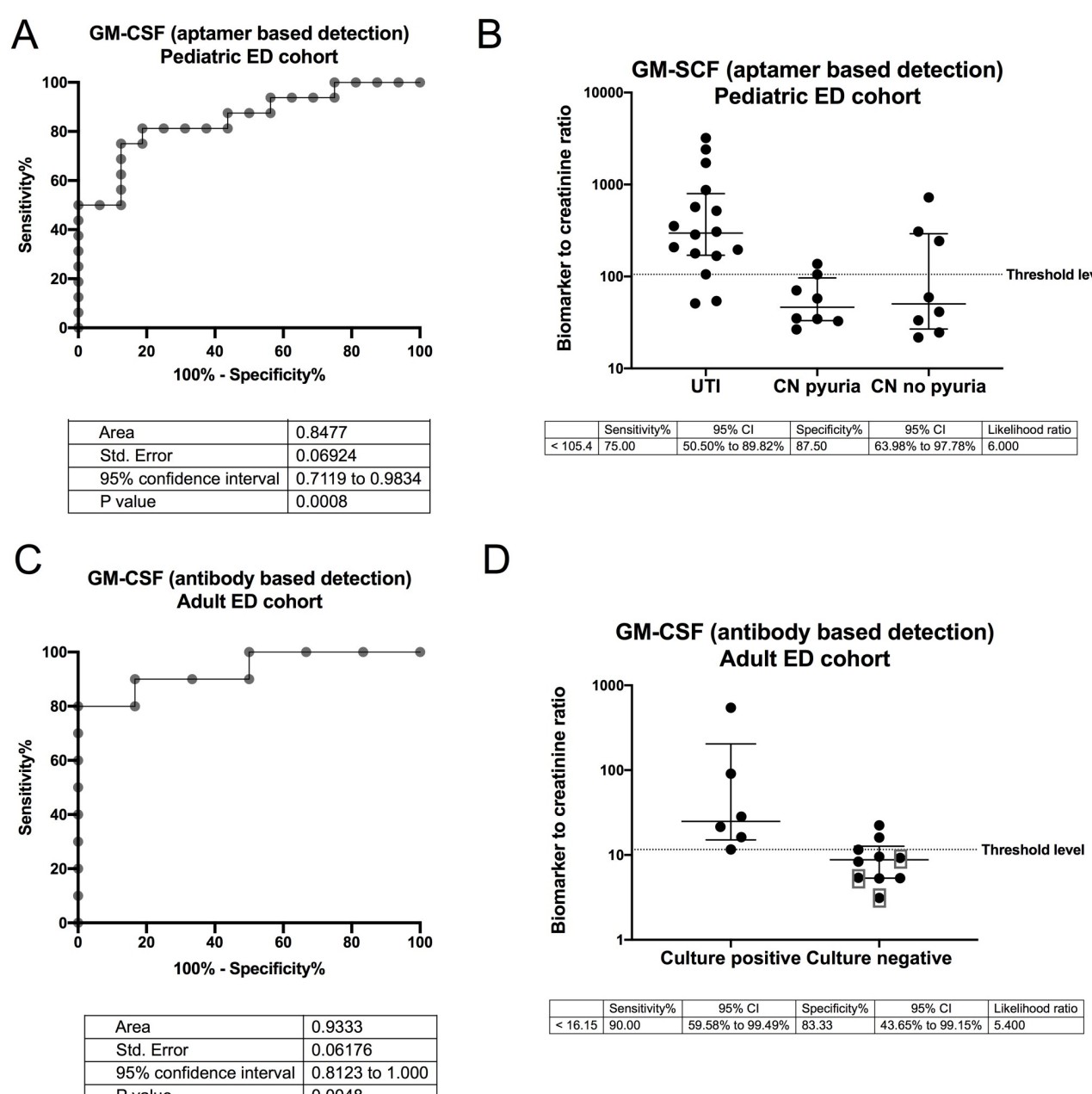

**Fig 3.** Scatter plots for GM-CSF measured adult ED urine samples using an antibody based Mesoscale platform (A) and measured in pediatric ED urine samples using an aptamer based Somalogic platform (B) are presented. AUCs for the antibody based (C) and the aptamer-based platforms were similar. Units for aptamer-based detection are RFU of urine GM-CSF per mg of urine creatinine and units for antibody based detection are pg of urine GM-CSF per mg of urine creatinine. The patients were selected from an adult population from the Ohio State University Wexner Medical Center Emergency Department and included 6 culture positive and 10 culture negative samples. The mean (range) ages in years were 74 (65–89) for the culture positive and 62 (47–74) for the culture negative group. The majority, 4/6 (67%) in the culture positive and 8/10 (80%) in the culture negative group were female. Urine cultures grew E. coli in four patients, E. Faecalis in one patient and both E. coli and E. Faecalis in one patient. Seven of the culture negative group had no growth and three represented by gray boxes grew mixed flora. Of note these results may not differentiate UTI from asymptomatic bacteriuria which will need investigated in future studies.

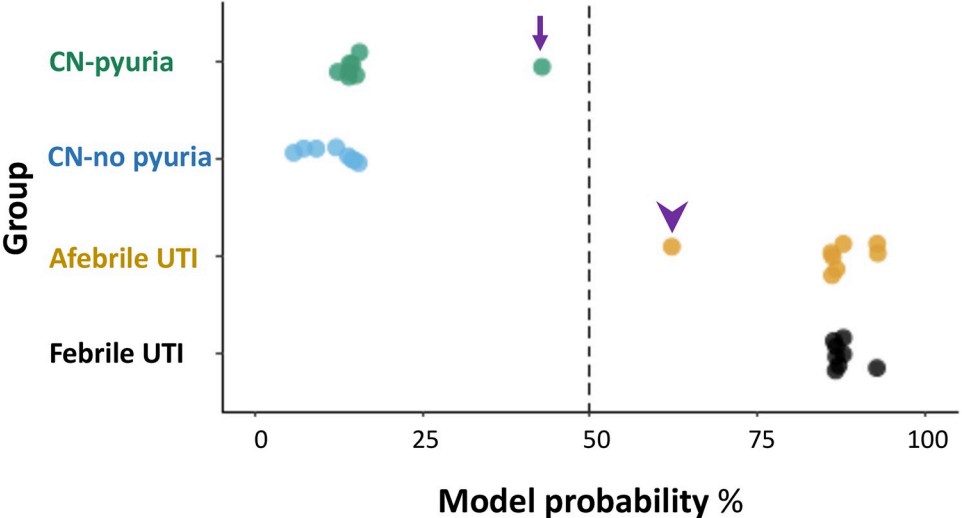

**Fig 4. The UTI class probability estimate for each sample by the optimal SVM classifier.** The dashed black line shows where the 50% probability lies. Generally, model probability of predicting UTI samples was > 80%. There are 2 outliers, one 18-year old female with CN pyuria (purple arrow) who presented with left flank pain, fever and dysuria along with 1+ LE on UA and had 43.4% UTI probability. The other outlier (purple arrowhead) was a 3-year-old female who presented with fever and abdominal pain, along with 1+ LE and had 62.7% UTI probability.

selection during the 5-fold cross-validation process are shown in Fig 4A and S4 Material. The UTI class probability estimate calculated based on the SVM decision values for each sample are presented as Fig 4B.

## Source material

The source data is presented as S5 Material.

## Discussion

Analysis of the human urine is the 1st known type of laboratory medicine and dates back to 4,000 BCE [24]. Hippocrates (460–355 BCE) associated increasing urine sediment with increasing fevers [24]. If the aforementioned urine sediment was due to WBCs this would be the earliest known description of a UTI biomarker [24]. Urine test strips, sometimes referred to urine dipsticks were developed in the 1950s and 1960s and have been used since then in the point-of-care diagnosis of UTIs [25]. However, UAs have limitations regarding sensitivity and specificity and its key UTI diagnostic components detects LE in the urine, a finding not necessarily specific to UTIs. Despite the need for more judicious use of antibiotics secondary to increasing rates of antibiotic resistant bacteria, point of care diagnosis of UTIs has remained largely unchanged since the introduction of urine dipsticks. One strategy is to detect bacterial products such as bacterial nuclease activity, however identifying the bacterial load indicative of a UTI may be problematic because urine contains a microbiota [26]. We and others have identified increased urine levels of innate immune proteins in the urine compared to normal controls [27, 28]. However, these innate immune proteins might not associate with UTIs if compared to the urine of ill patients without UTI. Here we use patients for which a urine dipstick and culture was obtained for clinical indications highlighting advantages of an emergence department for biomarker studies. Specifically, we identified a protein profile that differentiates UTI from children who had similar symptoms but did not have a UTI (e.g. CN-no pyuria and CN-pyuria).

We identified 8 proteins that were significantly (p < 0.01) elevated in UTI samples compared to CN-pyuria and CN-no pyuria samples with "excellent" biomarker potential based on an AUC of 0.9–1 (Fig 1). Some of the candidate proteins (Fig 1) have been associated with bacterial interactions with mucosal surfaces other than the urinary tract. Cathepsin S (CTSS) expression is upregulated during periodontal infections [29]. Transgelin 2 mimics bacterial SipA, a protein that promotes bacterial entry into cells, and promotes phagocytosis in lipopolysaccharide activated macrophages [30]. C-X-C motif chemokine 13 is required for recruitment of specialized B cells, antibody production and the bacterial defense of the peritoneal and pleural cavities [31]. The involvement of cathepsin S, transgelin 2 and C-X-C motif chemokine 13 with other infections provides a foundation for the evaluation of the potential role of these proteins in *E.coli* UTI pathophysiology while other proteins have been linked to viral infections. B cell lymphoma protein 6 was initially described for its regulation of lymphocyte growth and development, but has been demonstrated to function as a checkpoint regarding the initiation of the innate immune response to cytosolic RNA viruses [32]. Mitogen protein kinase 9 and Protein E7 HPV18 are also involved in innate response to viral infections [33, 34]. Past studies of the human virome have had variable results regarding increased Protein E7 HPV18 expression during UTI [35, 36]. HPV18 is included in the vaccine for this virus, however 24/32 (75%) of included patients were < 11 years of age, younger than the recommend age for the HPV vaccine [37]. It is possible that Protein E7 HPV18 represents a virus with homologous regions such as adenovirus E1a [38]. Other proteins not included in our top 8 candidate biomarkers were significantly elevated in the UTI group, but had AUCs < 0.9 (Table 1). In some cases, these proteins have more established roles in UTI pathophysiology. Pulmonary surfactant-associated protein D (SP-D) inhibits the growth of uropathogenic *E.coli* and regulates renal inflammation via the p38 MAPK related pathway during UTI [39]. Granulocyte macrophage colony-stimulating factor (GM-CSF) has been shown to be expressed by murine urothelial cells in response to lipopolysaccharides [40]. Our findings provide further biological relevance for SP-D and GM-CSF in human UTIs. We speculate that a panel containing some of the aforementioned biomarkers might lead to improved UTI diagnosis compared to urine dipstick results. Further we obtained similar results using aptamer and antibody-based GM

Current urine dipstick markers of host immune response (e.g. LE) are limited to proteins produced by WBCs, and thus may be produced nonspecifically when WBCs are present in the urine. In our previously reported larger cohort of 199 patients that we selected our samples from for this study, had a sensitivity of 83% and specificity of 85%. Other pediatric meta-analyses/studies reported sensitivities of 72–83% and specificities of 78–87% for LE. In comparison B-cell lymphoma protein and mitogen activated protein kinase 9 each had a sensitivity of 88% and specificity of 94%. Immunohistochemistry images for from the Human Protein Atlas version 18.1 (https://www.proteinatlas.org) are consistent with mitogen activated protein 5, along with C-X-C chemokine ligand 13 and heat shock protein are expression in the spleen, bladder lumen and collecting duct of the kidney [41–43]. We have previously demonstrated that the renal collecting duct, the initial kidney tubular section encountered by ascending uropathogens has innate immune functions [44]. Since many of the biomarkers that we evaluated are expressed in the bladder and kidney, in addition to the spleen (e.g. myeloid cells), they may represent a more specific innate immune response to infection compared to white blood cell limited proteins such as the LE.

To the best of our knowledge, this is the first study that uses Somacan™ proteomics data to construct a machine learning predictive model for urinary tract infection. In this study, we explored the application of SVM classifier in solving the classification problem on proteomics data. To obtain an unbiased performance estimation, we have adopted a nested cross-validation approach that performing hyperparameter tuning and model optimization in the inner

cross-validation loop and evaluated the optimal models independently in the outer cross-validation loop [45]. This design avoids the optimistic bias introduced into the performance estimate due to the use of the same cross-validation procedure for both hyperparameter optimization and performance evaluation [46]. Our SVM model had a slightly lower AUC than for some of the individual proteins. This is likely because for the SVM model that divided our results between a test and validation cohort. We anticipate that the SVM model will outperform single biomarkers in future studies with many more samples. It is also possible that our SVM model may be more accurate than urine culture. The patient that was assigned to the CN pyuria with a UTI probability of score of 43.4% (Fig 4) presented with left flank pain, fever, dysuria and UTI history; we speculate that this patient might have had an actual UTI with an organism that could not be isolated in standard urine. In the future, enhanced urine culture and sequencing could be applied to similar samples to help determine if these actually represent culture negative UTIs [7].

There are limitations with this study. First, we did not attempt to identify a biomarker to distinguish between pyelonephritis and cystitis in this study because we did not have radiologic evidence to distinguish between these conditions. Second, because of the expense of analyzing >1300 proteins rather than a targeted study comparing 1–10 proteins, a relatively small number of urine samples, 32, were analyzed and stringent filtering criteria such as using a p value of 0.1 rather than 0.05 and an AUC of 0.9 rather than 0.8. With this strict criterion we may have excluded some relevant biomarkers. Third, the Somascan™ platform does not include all proteins. For example, human alpha defensin 5 and human neutrophil proteins 1–3 which we reported on in 2016 in a hypothesis based UTI biomarker study are not included in the Somascan™ platform [17]. Fourth, the Somascan™ platform reports results a relative fluorescent units (RFU/ml); use of additional methodologies such as ELISA will be needed for further quantification of the candidate proteins. Fifth, future studies will be needed to see if different populations such as children less than 2 years of age, patients with urine collected by catheter or patients with positive urine cultures and no LE have different urine biomarker profiles. Last, we limited our pediatric UTI samples to cultures from which *E.coli*, the bacteria species that accounts for 75%-90% of UTIs, was isolated, it is possible that other bacteria species may result in distinct proteomic patterns [47].

LE is a nonspecific marker for pyuria while nitrites are a bacterial product. LE may not differentiate culture negative pyuria from UTI and nitrites might not differentiate asymptomatic bacteriuria from UTI. Addition of urine levels of renal urothelium and collecting duct expressed innate immune proteins adds a distinct biomarker mechanism to current UTI diagnosis methodology. Our 8 leading proteins had AUCs between 0.91 and 0.95 indicating that they represent "excellent" biomarker candidates [20]. We propose that including the urine levels of these candidate biomarkers, either alone or in combination with traditional UA biomarkers will help clinicians identify true UTI from culture negative pyuria at the point of care. Future directions will be testing a larger number of patients with ELISAs to further refine our SVM model. Ultimately any biomarker panel will need to be converted to an aptamer-based point of care test and validated with an antibody-based assay, similar to what was done with GM-CSF in this study.

## Supporting information

**S1 Material. Primers used for RT-PCR.**
(PDF)

**S2 Material. Illustration of machine learning process using a nested cross-validation strategy.** The support vector machine (SVM) models are trained and selected in an inner leave-

one-out cross-validation loop, which involves model-based feature selection and SVM hyper-parameter optimization. An outer 5x cross-validation loop evaluates the generalized performance of the optimal model selected from the inner loop on the test set.
(PDF)

**S3 Material. Table of proteins that were statistically different in UTI vs CN no pyuria urine and UTI versus CN pyuria with a p value of $< 0.05$ but were not statistically different between CN pyuria and CN no pyuria samples.**
(XLSX)

**S4 Material. Expression patterns of the 17 proteins selected by random forest algorithm presented as heatmap with UTI versus non-UTI (CN pyuria and CN no pyuria).**
(PDF)

**S5 Material. Source data.**
(XLS)

## Acknowledgments

We would like to acknowledge Jennifer Kline (Nationwide Children's) and Michael Hill (The Ohio State University) for coordinating sample collection.

## Author Contributions

**Conceptualization:** Joshua Watson, David S. Hains, Andrew L. Schwaderer.

**Data curation:** Joshua Watson, Samuel Arregui, John Ketz.

**Formal analysis:** Liang Dong, Sha Cao, Jeffrey M. Caterino, David S. Hains, Andrew L. Schwaderer.

**Funding acquisition:** Andrew L. Schwaderer.

**Investigation:** Samuel Arregui, Vijay Saxena, Abduselam K. Awol, David S. Hains, Andrew L. Schwaderer.

**Methodology:** Liang Dong, Vijay Saxena, Andrew L. Schwaderer.

**Project administration:** Andrew L. Schwaderer.

**Resources:** Andrew L. Schwaderer.

**Supervision:** Joshua Watson, Daniel M. Cohen.

**Writing – original draft:** Liang Dong, Andrew L. Schwaderer.

**Writing – review & editing:** Joshua Watson, Sha Cao, Samuel Arregui, Vijay Saxena, John Ketz, Abduselam K. Awol, Daniel M. Cohen, Jeffrey M. Caterino, David S. Hains, Andrew L. Schwaderer.

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
