## [Decision Letter · Decision Letter 0]

13 Jan 2020

PONE-D-19-29587

Aptamer based proteomic pilot study reveals a urine signature indicative of pediatric urinary tract infections.

PLOS ONE

Dear Dr Schwaderer,

Thank you for submitting your manuscript to PLOS ONE. After careful consideration, we feel that it has merit but does not fully meet PLOS ONE’s publication criteria as it currently stands. Therefore, we invite you to submit a revised version of the manuscript that addresses the points raised during the review process.

We would appreciate receiving your revised manuscript by Feb 27 2020 11:59PM. To enhance the reproducibility of your results, we recommend that if applicable you deposit your laboratory protocols in protocols.io, where a protocol can be assigned its own identifier (DOI) such that it can be cited independently in the future. For instructions see: http://journals.plos.org/plosone/s/submission-guidelines#loc-laboratory-protocols

We look forward to receiving your revised manuscript.

Kind regards,

Mehreen Arshad, M.D.

Academic Editor

PLOS ONE

Journal Requirements:

'The work was funded by Lilly Endowment, Inc. Physician Scientist Initiative (ALS and

DSH) and Nationwide Children’s Hospital intramural funds (ALS and JW). ALS JMC and DSH

were supported by the National Institute on Aging R01AF050801 (JC, ALS and DSH). ALS and

DSH were supported in part by grants from the United States National Institute of Diabetes and

Digestive and Kidney Diseases (R01DK106286 and R01DK117934)'

'Eli Lily foundation award (https://www.lilly.com/who-we-are/lilly-foundation) ALS and DSH

The Research Institute at Nationwide Hospital intramural funds (https://www.nationwidechildrens.org/research). ALS

Grant numbers: NA

The sponsors played no role in study design, data collection and analysis, decision to publish, or preparation of the manuscript'

Please provide an amended Funding Statement that declares *all* the funding or sources of support received during this specific study (whether external or internal to your organization) as detailed online in our guide for authors at http://journals.plos.org/plosone/s/submit-nowPlease state what role the funders took in the study.  If any authors received a salary from any of your funders, please state which authors and which funder. If the funders had no role, please state: "The funders had no role in study design, data collection and analysis, decision to publish, or preparation of the manuscript."

'ALS has consulted for Allena Pharmaceuticals on a topic unrelated to this manuscript.  Others there are no competing interests'

Reviewers' comments:

Reviewer's Responses to Questions

**Comments to the Author**

1. Is the manuscript technically sound, and do the data support the conclusions?

Reviewer #1: Yes

Reviewer #2: Yes

2. Has the statistical analysis been performed appropriately and rigorously? 

Reviewer #1: Yes

Reviewer #2: Yes

3. Have the authors made all data underlying the findings in their manuscript fully available?

Reviewer #1: Yes

Reviewer #2: Yes

4. Is the manuscript presented in an intelligible fashion and written in standard English?

Reviewer #1: Yes

Reviewer #2: Yes

5. Review Comments to the Author

Reviewer #1: The authors have provided evidence to suggest that there are compositional differences in the urinary proteome of individuals with UTI and pyuria and those with pyuria but no evidence of UTI. Several of the makers appear to be unique in this context. While many markers appear to provide high sensitivity their specificity is quite often low. This last point is truly difficult to assess because of the low numbers of patients examined.

Since the use of dipstix and esterase monitoring was so heavily criticized and it is the current standard it would have been useful to include data regarding the results from the assay to compare performance. The manuscript is comparing a very expensive assay to a relatively inexpensive one. Clearly the intent would be to hone the panel down to a much smaller set of analytes but the comparison might have further strengthened the authors case.

I question the value/point of providing the three sets of machine learning results as they are quite disparate and there is no significant discussion of the relative merits or limitations of these approaches. Thus it is unclear what this data adds to the current document.

I found the reporting of fluorescence intensity rather confusing as I am familiar with significantly higher signals with somascan than were listed. This may arise from the normalization to creatinine levels. It would be beneficial to determine if the creatinine levels between each group were significantly different. The data to do this is available in the supplementary tables.

I failed to see the value of the inclusion of supplementary figures 6 and 7 as this could just as easily been stated in the discussion. The data derives from protein atlas and has no relationship to the current study other than saying there is evidence to suggest the presence of the candidate proteins in renal and bladder tissues. Similarly I do not see the need/utility of the PCR data on unrelated and incompletely rationalized samples from different tissue sources.

Somascan is very robust and sensitive with a very board dynamic range. It is essential to have some form of validation using an antibody based approach to demonstrate that ELISA is actually feasible for these analytes at the concentrations present in urine.

Reviewer #2: This paper by Dong et al. presents an interesting work and potentially holds very promising impact on POC diagnostics for UTI, and I found this work interesting. However, there are few queries and suggestions that need to be addressed to improve the quality of manuscript.

Comment 1: Language at times is bit difficult to understand, particularly in the paragraph 2-5 of the discussion, the linkage between the paragraphs are also missing.

Comment 2: Similarly, in the introduction many lines are lengthy which can be difficult to comprehend for readers, should be broken into smaller sentences.

For instance, consider this line

“Infectious conditions including renal tuberculosis, herpes simplex virus and chlamydia along

with noninfectious inflammatory conditions including Kawasaki disease, appendicitis, foreign

bodies and interstitial nephritis can cause what has been historically described as sterile pyuria”

Comment 3: Proper referencing is missing at times for instance, consider

“Several challenges have limited protein-based biomarker discovery including the difficulty to develop high throughput assays such as enzyme-linked immunosorbent assay (ELISA) for identifying biomarker candidates.”

Comment 4: The problems with existing technologies, like ELISA should be briefly discussed in the introduction and how aptamer-based technologies can alleviate them should also be discussed in brief with proper references.

Comment 5: It would be good to include some information on the current potential and market of aptamers in Introduction. Some the recent publications on this aspect is as follows.

1- Aptamer-based point-of-care diagnostic platforms

2- Aptamers in the Therapeutics and Diagnostics Pipelines

3- The point behind translation of aptamers for point of care diagnostics

4- Defining Target Product Profiles (TPPs) for Aptamer-Based Diagnostics

6. PLOS authors have the option to publish the peer review history of their article (what does this mean?). If published, this will include your full peer review and any attached files.

Reviewer #1: Yes: John A Wilkins

Reviewer #2: No

---

## [Author Response · Author response to Decision Letter 0]

13 May 2020

We have revised the manuscript based on the reviewers comments resulting in a more concise and focused study. We have also performed additional studies when recommended. Please see below for an itemized response. 

Since the use of dipstix and esterase monitoring was so heavily criticized and it is the current standard it would have been useful to include data regarding the results from the assay to compare performance. The manuscript is comparing a very expensive assay to a relatively inexpensive one. Clearly the intent would be to hone the panel down to a much smaller set of analytes but the comparison might have further strengthened the authors case.

Response: For this study we purposely chose 16 patients with a positive culture and positive leukocyte esterase (LE) on dipstick, 8 with a negative culture and positive LE and 8 with negative LE and culture. Therefore LE was predetermined to have a sensitivity of 50% and specificity of 33%. In our previously reported larger cohort of 199 patients that we selected our samples from LE >/= small had a sensitivity of 83% and specificity of 85% (PMID: 26885759). We have added this language and the reference to our discussion. 

I question the value/point of providing the three sets of machine learning results as they are quite disparate and there is no significant discussion of the relative merits or limitations of these approaches. Thus it is unclear what this data adds to the current document.

Response: We have removed the former figure 3 with three machine learning data sets and focused the results on the Random Forest algorithm. 

I found the reporting of fluorescence intensity rather confusing as I am familiar with significantly higher signals with somascan than were listed. This may arise from the normalization to creatinine levels. It would be beneficial to determine if the creatinine levels between each group were significantly different. The data to do this is available in the supplementary tables.

The urine creatinine levels were not different between groups, P = 0.523 using the Kruskal-Wallis test because the data was not parametric. Additionally we, in the initial submission, presented the ratio of urine biomarker to creatinine with the Somalogic results as relative fluorescent units (RFU) per ml and the creatinine as mg/dl for a dimensionally complex ratio of biomaker(RFU/ml)/ creatine (mg/dl). The higher creatinine volume (dl as opposed to ml) results in lower ratio values. We have revised Figure 2 and Table 2 with the creatinine results as mg/ml so mls cancel out and the results are the more traditional and simple biomarker (RFU)/creatinine (mg) units. 

Response:

I failed to see the value of the inclusion of supplementary figures 6 and 7 as this could just as easily been stated in the discussion. The data derives from protein atlas and has no relationship to the current study other than saying there is evidence to suggest the presence of the candidate proteins in renal and bladder tissues. Similarly I do not see the need/utility of the PCR data on unrelated and incompletely rationalized samples from different tissue sources.

Response: We have removed supplementary figures 6 and 7 and instead added the presence of candidate biomarkers in renal and bladder tissue to the discussion as recommended.

Somascan is very robust and sensitive with a very board dynamic range. It is essential to have some form of validation using an antibody based approach to demonstrate that ELISA is actually feasible for these analytes at the concentrations present in urine.

Response: We have validated the Somalogic results for GM-CSF with an antibody based approach in a separate patient cohort (an adult patient cohort) which resulted in a similar area under the curve value. The results are presented in a new Figure 3. We have also updated the methods, results and discussion based on this finding.

Comment 1: Language at times is bit difficult to understand, particularly in the paragraph 2-5 of the discussion, the linkage between the paragraphs are also missing.

Response: We have revised the wording and added linkage for paragraphs 2-5

Comment 2: Similarly, in the introduction many lines are lengthy which can be difficult to comprehend for readers, should be broken into smaller sentences.

For instance, consider this line:

“Infectious conditions including renal tuberculosis, herpes simplex virus and chlamydia along with noninfectious inflammatory conditions including Kawasaki disease, appendicitis, foreign bodies and interstitial nephritis can cause what has been historically described as sterile pyuria”

Response: We have shortened sentences and simplified the introduction as recommended. 

Comment 3: Proper referencing is missing at times for instance, consider

“Several challenges have limited protein-based biomarker discovery including the difficulty to develop high throughput assays such as enzyme-linked immunosorbent assay (ELISA) for identifying biomarker candidates.”

Response: We have add references for the example cited and in other places as well. 

Comment 4: The problems with existing technologies, like ELISA should be briefly discussed in the introduction and how aptamer-based technologies can alleviate them should also be discussed in brief with proper references.

Response: We have provided an extended explanation for ELISA limitation in the introduction: “Several challenges have limited protein-based biomarker discovery with traditional antibody based ELISAs. Specifically, ELISAs are time consuming to perform and the required antibodies have inherent costs, instability, batch to batch variation, storage requirements limited dynamic ranges and are difficult to multiplex [10-12].”

Comment 5: It would be good to include some information on the current potential and market of aptamers in Introduction. Some the recent publications on this aspect is as follows.

1- Aptamer-based point-of-care diagnostic platforms

2- Aptamers in the Therapeutics and Diagnostics Pipelines

3- The point behind translation of aptamers for point of care diagnostics

4- Defining Target Product Profiles (TPPs) for Aptamer-Based Diagnostics

Response: We have added more information about the current and market potential of aptamers in the introduction: Aptamers are being explored as affordable, sensitive, specific, user friendly point of care tests on a variety of platforms [15]. Aptamers have the ability to perform in formats where antibodies often perform poorly such as homogenous multiplex assays, do not degrade when stored at room temperature as a dry lyophilized reagent at room temperature and have minimal to no batch to batch variation [12]. Thus there is speculation that aptamers may replace antibodies in future diagnostics [16]”

---

## [Editor Report · Decision Letter 1]

15 Jun 2020

Aptamer based proteomic pilot study reveals a urine signature indicative of pediatric urinary tract infections.

PONE-D-19-29587R1

Dear Dr. Schwaderer,

We’re pleased to inform you that your manuscript has been judged scientifically suitable for publication and will be formally accepted for publication once it meets all outstanding technical requirements.

Kind regards,

Mehreen Arshad, M.D.

Academic Editor

PLOS ONE

---

## [Editor Report · Acceptance letter]

23 Jun 2020

PONE-D-19-29587R1 

Aptamer based proteomic pilot study reveals a urine signature indicative of pediatric urinary tract infections. 

Dear Dr. Schwaderer:

I'm pleased to inform you that your manuscript has been deemed suitable for publication in PLOS ONE. Congratulations! Your manuscript is now with our production department. 

Kind regards, 

on behalf of

Dr. Mehreen Arshad 

Academic Editor

PLOS ONE